# Astrocytic modulation of population encoding in mouse visual cortex via GABA transporter 3 revealed by multiplexed CRISPR/Cas9 gene editing

Jiho Park[1,2†], Grayson O Sipe[3†], Xin Tang[4,5], Prachi Ojha[2], Giselle Fernandes[2], Yi Ning Leow[1,2], Caroline Zhang[1], Yuma Osako[2], Arundhati Natesan[2], Gabrielle T Drummond[1,2], Rudolf Jaenisch[6], Mriganka Sur[1,2]*

[1]Department of Brain and Cognitive Sciences, Massachusetts Institute of Technology, Cambridge, United States; [2]Picower Institute for Learning and Memory, Massachusetts Institute of Technology, Cambridge, United States; [3]Department of Biology, Eberly College of Science and Huck Institutes of the Life Sciences, Pennsylvania State University, University Park, United States; [4]F.M. Kirby Neurobiology Center, Boston Children's Hospital, Harvard Medical School, Boston, United States; [5]Department of Neurosurgery, Boston Children's Hospital, Boston, United States; [6]Whitehead Institute for Biomedical Research, Cambridge, United States

*For correspondence: msur@mit.edu

†These authors contributed equally to this work

Competing interest: The authors declare that no competing interests exist.

## eLife Assessment

In this manuscript, Park et al. developed a multiplexed CRISPR construct to genetically ablate the GABA transporter GAT3 in the mouse visual cortex, with effects on population-level neuronal activity. This work is **important**, as it sheds light on how GAT3 controls the processing of visual information. The findings are **compelling**, leveraging state-of-the-art gene CRISPR/Cas9, in vivo two-photon laser scanning microscopy, and advanced statistical modeling.

**Abstract** Astrocytes, which are increasingly recognized as pivotal constituents of brain circuits governing a wide range of functions, express GABA transporter 3 (Gat3), an astrocyte-specific GABA transporter responsible for maintenance of extra-synaptic GABA levels. Here, we examined the functional role of Gat3 in astrocyte-mediated modulation of neuronal activity and information encoding. First, we developed a multiplexed CRISPR construct applicable for effective genetic ablation of Gat3 in the visual cortex of adult mice. Using in vivo two-photon calcium imaging of visual cortex neurons in Gat3 knockout mice, we observed changes in spontaneous and visually driven single neuronal response properties such as response magnitudes and trial-to-trial variability. Gat3 knockout exerted a pronounced influence on population-level neuronal activity, altering the response dynamics of neuronal populations and impairing their ability to accurately represent stimulus information. These findings demonstrate that Gat3 in astrocytes profoundly shapes the sensory information encoding capacity of neurons and networks within the visual cortex.

## Introduction

Astrocytes, once thought to provide only passive support to neurons, are now recognized as active modulators of neuronal activity and behavior. Many recent studies have demonstrated that astrocytes significantly influence neuronal activity via diverse mechanisms that include regulation of the release and clearance of neurotransmitters and other neuroactive molecules at synaptic and extra-synaptic sites (*Santello et al., 2019*; *Perea et al., 2014*; *Volterra and Meldolesi, 2005*; *Miguel-Quesada et al., 2023*; *Khakh and Sofroniew, 2015*; *Haim and Rowitch, 2017*; *Verkhratsky and Nedergaard, 2018*). A notable mechanism through which astrocytes could impact cortical dynamics is by modulating inhibitory transmission, as astrocytes express a rich repertoire of GABA-related proteins that enable them to synthesize, release, and clear GABA (*Liu et al., 2022*; *Kwak et al., 2020*). They are often found spatially co-localized with GABAergic synapses, placing them in a position to regulate inhibitory synapses and influence local synaptic transmission, and thus network dynamics and behavior (*Mederos et al., 2021*; *Perea et al., 2016*). However, the specific effects of astrocytic modulation of inhibitory signaling on functional neuronal circuits remain unknown.

An important yet understudied mechanism of astrocytic regulation of inhibitory transmission is GABA uptake via the astrocyte-specific GABA transporter 3 (Gat3; *Slc6a11*). Unlike GAT1 (*Slc6a1*), which is primarily expressed by neurons at synaptic sites and facilitates rapid clearance of synaptic GABA (*Melone et al., 2014*; *Kersanté et al., 2013*; *Song et al., 2013*; *Scimemi, 2014*), Gat3 is exclusively expressed in astrocytic processes and speculated to regulate ambient tonic, rather than phasic, levels of GABA in the extra-synaptic space (*Song et al., 2013*; *Minelli et al., 1996*). Tonic inhibition mediated by extra-synaptic GABA receptors is important for modulating the gain and maintaining the tone of neuronal activity via regulating neuronal excitability (*Koh et al., 2023*; *Semyanov et al., 2004*; *Lee and Maguire, 2014*). Much of our current understanding of Gat3 function has been limited to pharmacological studies performed *ex vivo*. The findings from such studies have shown that Gat3 can significantly influence single neuron properties, observed as changes in extracellular GABA levels and inhibitory postsynaptic currents (IPSCs) in a cell-type-specific and state-dependent manner (*Kersanté et al., 2013*; *Matos et al., 2018*; *Héja et al., 2012*; *Boddum et al., 2016*; *Shigetomi et al., 2012*).

Gat3 has been implicated in several neurological dysfunctions: reduction of Gat3 in the thalamus due to reactive astrogliosis leads to neuronal hyperexcitability and an increased risk of seizures (*Cho et al., 2022*); increased Gat3 activity in the striatum underlies excessive self-grooming behavior in mice (*Yu et al., 2018*); and Gat3 reduction in the globus pallidus impairs motor coordination (*Kang et al., 2023*; *Chazalon et al., 2018*). Despite these findings, our understanding of the role of Gat3 in the cerebral cortex remains limited and even obscured by somewhat contradictory findings. For example, two studies have yielded contrasting results, with one study showing that pharmacological manipulation of Gat3 increases interneuron excitability (*Kinney, 2005*) but another showing that it has no effect on synaptic transmission or cell excitability (*Keros and Hablitz, 2005*). These findings emphasize the regional specificity of Gat3 function and its importance in maintaining proper circuit dynamics.

A fundamental challenge in resolving Gat3's circuit function *in vivo* has been the lack of tools to achieve precise manipulation of Gat3. Pharmacological agents targeting Gat3, such as SNAP-5114, often have off-target effects and, particularly when administered systemically, can introduce confounding variables due to macroscale circuit changes and adaptations. Conventional transgenic approaches are complicated by developmental confounds and adaptations in circuit development that can obscure the acute impact of Gat3 on network function (*Tang et al., 2021*). To address these limitations, we developed a multiplexed CRISPR construct that allows *in vivo* delivery of multiple CRISPR knockout sgRNAs (MRCUTS: Multiple sgRNA Csy4-mediated Universal Targeting System) to effectively ablate Gat3 in the cortex. While our tool is not inherently cell-type specific, it can be used with commercially available mouse lines with conditional Cas9 overexpression for rapid and efficient cell-type-specific knockout of genes in adult animals. Here, because of the specificity of Gat3 expression pattern in astrocytes (*Melone et al., 2014*; *Scimemi, 2014*; *Minelli et al., 1996*), we achieved astrocyte-specific Gat3 knockout using Cas9 expression driven by a ubiquitous promoter.

This tool enabled us to perform the first *in vivo* investigation of astrocyte Gat3 function at both single neuron and population levels in the cortex. We hypothesized that astrocytes organize neuronal activity in the cortex by regulating tonic inhibition via Gat3 to optimize information encoding in cortical circuits. To test this hypothesis, we assessed the impact of astrocyte Gat3 ablation using *in vivo* two-photon

imaging to capture neuronal activity in the mouse visual cortex. The visual cortex represents an ideal system to investigate this question, as visual cortical neurons have well-characterized response properties and established paradigms for assessing both single neuron and population-level information encoding—properties known to be sculpted by both tonic and phasic inhibitory transmission (*Priebe, 2016*; *Pfeffer et al., 2013*; *Shapley et al., 2003*) which could be modulated by Gat3.

Our findings demonstrate that astrocytic Gat3 profoundly influences visual information processing on multiple scales—from modulating the response properties of individual neurons to orchestrating population-level representations of visual stimuli. These results establish astrocytes as active participants in cortical information processing and provide insight into how disruptions in astrocytic GABA transport may contribute to cortical circuit dysfunction in neurological disorders.

## Results
### Gat3 function can be studied *in vivo* with a single multiplexed CRISPR construct

Previous studies have shown that Gat3 is expressed across the brain: it is found in all cortical layers, as well as in subcortical structures such as the thalamus and the hypothalamus (*Pow et al., 2005*). We first examined the expression of Gat3 in the primary visual cortex (V1). Immunohistochemical staining of adult mouse V1 revealed that Gat3 expression was found throughout the visual cortex (*Figure 1A*). Consistent with earlier findings, we observed expression across all cortical layers, with enriched Gat3 expression especially in layers 2/3 to 5 (*Melone et al., 2014*; *Scimemi, 2014*; *Minelli et al., 1996*; *Figure 1A, B*).

Our initial pharmacological experiments using the Gat3 antagonist SNAP-5114 (*Figure 1—figure supplements 1 and 2*) revealed a subtle reduction in maximal responses in V1 neurons (*Figure 1—figure supplement 2B*), with no change in orientation selectivity index (OSI; *Figure 1—figure supplement 2D*). These subtle effects were highly variable, as with previous contradictory findings in the literature (*Kersanté et al., 2013*; *Song et al., 2013*; *Kinney, 2005*; *Keros and Hablitz, 2005*). These limitations of pharmacological approaches—including off-target effects, inability to achieve cell-type specificity, and challenges in differentiating direct effects from network adaptations—highlighted the need for a more precise genetic manipulation technique. To overcome these constraints, we developed a multiplexed CRISPR construct that would allow us to selectively ablate Gat3 in astrocytes with spatial and temporal precision *in vivo*.

We developed a multiplexed CRISPR/Cas9-based tool, Multiple sgRNA Csy4-mediated Universal Targeting System (MRCUTS), to selectively knock out one or more astrocytic genes with spatial and temporal selectivity. We opted to use a single AAV construct containing multiple CRISPR guide-RNA (gRNA) sequences delivered into transgenic mice expressing the Cas9 enzyme (*Ferreira et al., 2018*; *Kurata et al., 2018*; *Figure 1C*, see Methods). We first demonstrated that we could achieve effective knockout (KO) of Gat3 protein using the Gat3-MRCUTS construct *in vitro* in cultured astrocytes (*Figure 1D*). To evaluate the efficacy of Gat3 KO *in vivo*, we performed DNA sequencing from adult mouse brain tissue samples 4 weeks after the virus injection (C57BL/6J mice for control and CAG-Cas9-EGFP transgenic mice for Gat3 KO) (*Figure 1E*). In Gat3 KO samples, the gRNA-targeted areas showed a significantly higher level of genetic modifications, particularly deletions, in the Gat3 genomic region compared to control mice, which received the same sgRNA construct but did not express the Cas9 gene editing effector (*Figure 1E*). These experiments validated the efficacy of our CRISPR construct in knocking out Gat3 from astrocytes both *in vitro* and *in vivo*. Furthermore, *post hoc* immunohistochemistry of brain slices after two-photon imaging *in vivo* indicated that there was a significant reduction of Gat3 expression at the sites of injection in the Gat3 KO mice (*Figure 1F–H*). While there is high variability in Gat3 expression across hemispheres in control brain slices (*Figure 1G* and animal 2 in *Figure 1—figure supplement 3A*), the extent of Gat3 KO in Cas9+ mice was significantly greater than that in wild-type mice (*Figure 1G, H*, *Figure 1—figure supplement 3B*). These observations show that our multiplexed CRISPR construct reduced Gat3 expression with high efficacy. Importantly, our construct resulted in precise co-localization with neuronal jRGECO1a expression (*Figure 1G*), which validated our approach for studying Gat3 function *in vivo*.

Having confirmed successful Gat3 knockout at molecular and histological levels, we next sought to investigate how this manipulation would affect inhibitory transmission at a cellular level. To investigate

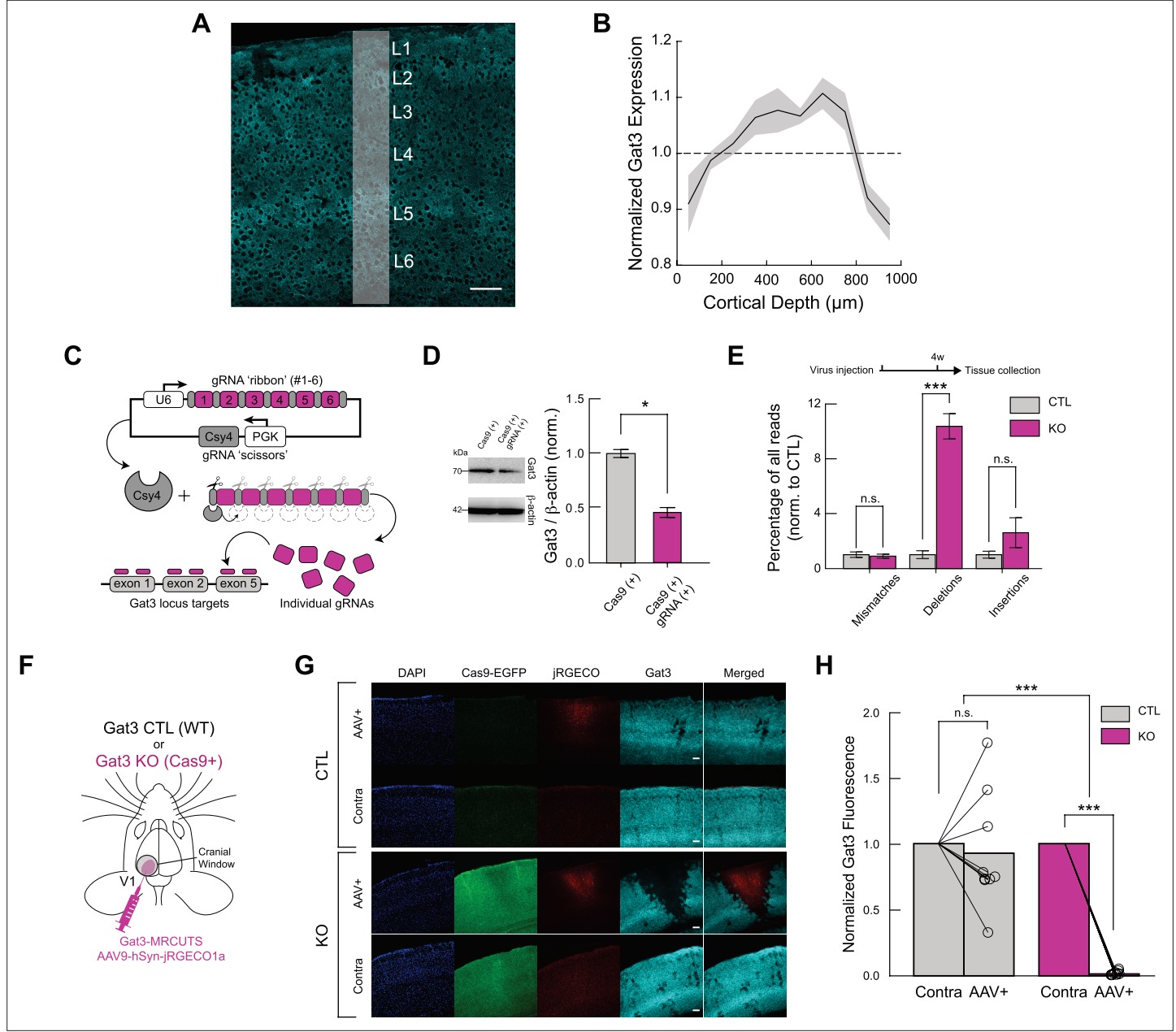

**Figure 1.** A Gat3-specific multiplexed CRISPR construct successfully knocks out Gat3. (**A**) Gat3 expression across cortical layers in the mouse visual cortex (scale bar = 100 μm). (**B**) Quantification of Gat3 expression across cortical layers; expression density in white strip is shown at right (*n* = 3 mice, shaded area = SEM). (**C**) Schematic diagram illustrating the construct design, which consists of six CRISPR KO sgRNAs targeting the mouse Gat3 gene. These sgRNAs are separated by Csy4 enzyme cleavage sites, allowing their individual release in virus-injected cells. (**D**) Western blot and its quantification showing efficient knockout of Gat3 in cultured astrocytes co-transfected with Cas9 and Gat3-MRCUTS plasmids compared to astrocytes transfected with Cas9 plasmid alone (*n* = 3 independent experiments, *p < 0.05, two-tailed unpaired *t*-test, error bars = SEM). (**E**) DNA sequencing reads of one gRNA targeted region from mouse brain tissue collected after virus injection shows frequency of deletions at the target site in KO tissue (n.s., $p_{mismatches}$ = 0.723; ***$p_{deletions}$ < 0.001; n.s., $p_{insertions}$ = 0.158, two-tailed unpaired *t*-test, error bars = SEM). (**F**) Schematic of viral injections and cranial window implant over V1 for two-photon imaging. Viral constructs of the multiplexed gRNAs and red-shifted calcium indicator were co-injected in the left hemisphere of either wild-type mice or Cas9-expressing transgenic mice. (**G**) Representative immunohistochemistry images from a control and KO animal (scale bar = 100 μm, applies to all images in a row). (**H**) Comparison of Gat3 fluorescence intensity at the imaging sites and at the non-injected site within individual slices. Baseline intensity was determined by the non-injected right (contralateral) hemisphere to account for variability between slices ($n_{control}$ = 9 slices, 4 mice, n.s., $p_{control}$ = 0.443; $n_{Gat3\ KO}$ = 10 slices, 4 mice, ***$p_{Gat3\ KO}$ < 0.001, Mann–Whitney *U* test, ***$p_{Between\ groups}$ < 0.001, two-way ANOVA).

The online version of this article includes the following figure supplement(s) for figure 1:

*Figure 1 continued on next page*

**Figure supplement 1.** Experimental design and timeline.

**Figure supplement 2.** Systemic administration of SNAP-5114 affects select V1 neuronal response properties.

**Figure supplement 3.** Gat3 expression changes in control and KO brain slices.

how Gat3 modulation can alter synaptic and tonic inhibition, we performed whole-cell patch clamp recordings of L2/3 pyramidal neurons in visual cortical slices, specifically examining IPSCs as a measure of GABAergic signaling alterations following Gat3 ablation (*Figure 2A*). In the Gat3 KO slices, pyramidal neurons had an increased spontaneous IPSC (sIPSC) frequency but not amplitude (*Figure 2B–E*). This observation indicated that Gat3 KO primarily affects presynaptic GABAergic signaling, likely through elevated ambient GABA that increases release probability, while preserving the postsynaptic response to individual vesicular release events. Our finding contrasts with previous work by *Kinney, 2005*, who reported increases in both frequency and amplitude of IPCSs in layer 5 pyramidal cells following SNAP-5114 application in rat brain slices. This discrepancy likely stems from experimental differences including layer-specific circuit properties (layer 2/3 vs. layer 5), species differences (mouse vs. rat), and manipulation approaches (genetic knockout vs. acute pharmacological inhibition). Despite these differences in synaptic phenotype details, our electrophysiological results, together with the molecular and histological evidence, conclusively demonstrate successful Gat3 knockout using our multiplexed CRISPR construct, with clear functional consequences at the cellular level.

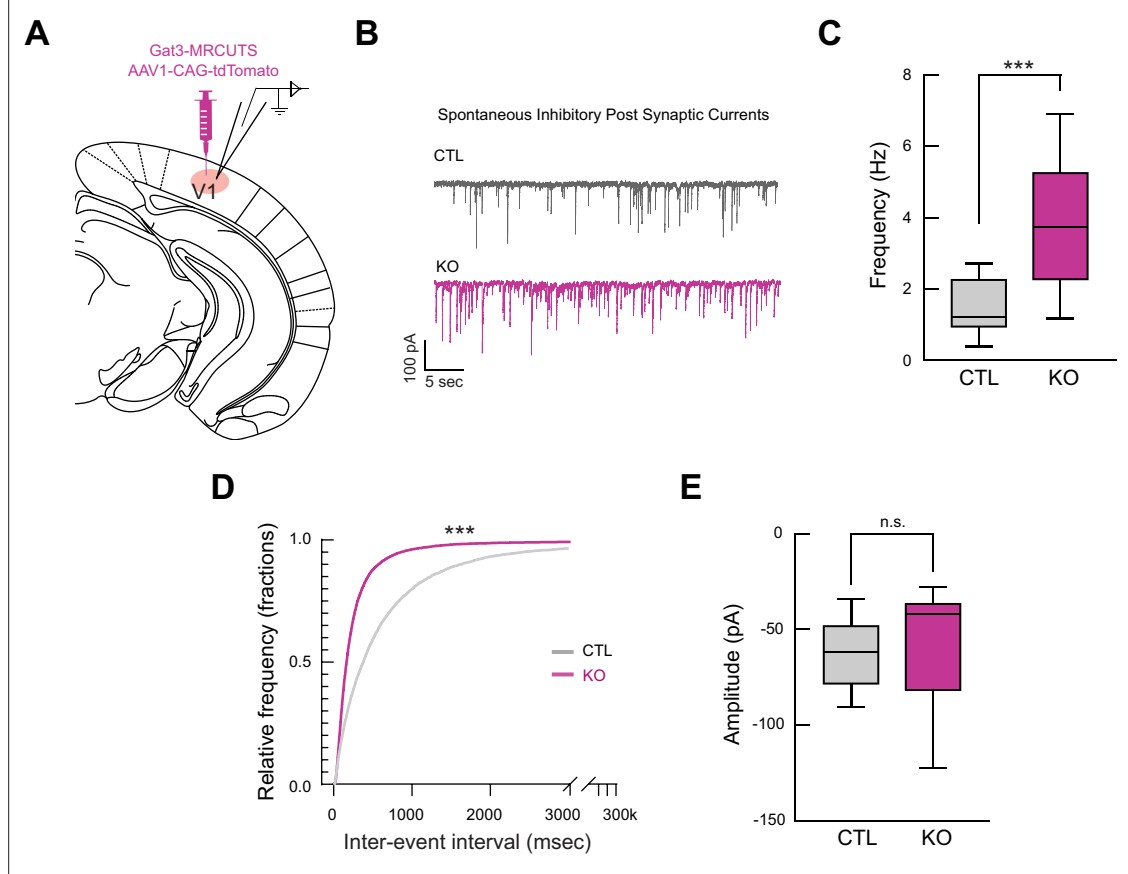

**Figure 2.** Genetic knockout of Gat3 in the visual cortex alters inhibitory output onto single pyramidal neurons. (**A**) Schematic of *ex vivo* whole-cell patch clamp electrophysiology setup. Gat3-MRCUTS was co-injected with a tdTomato virus to label the injection site for recordings. (**B**) Representative traces of spontaneous inhibitory postsynaptic currents (sIPSCs) of L2/3 pyramidal neurons in visual cortex brain slices. (**C**) Comparison of frequency of sIPSCs between control and Gat3 KO brain slices ($n_{control}$ = 20 cells, $n_{Gat3\ KO}$ = 23 cells, ***p < 0.001, two-tailed unpaired *t*-test). (**D**) Cumulative probability histograms for inter-event intervals (***p < 0.001, Kolmogorov–Smirnov test). (**E**) Comparison of average amplitude of sIPSCs (n.s., p = 0.9351, two-tailed unpaired *t*-test).

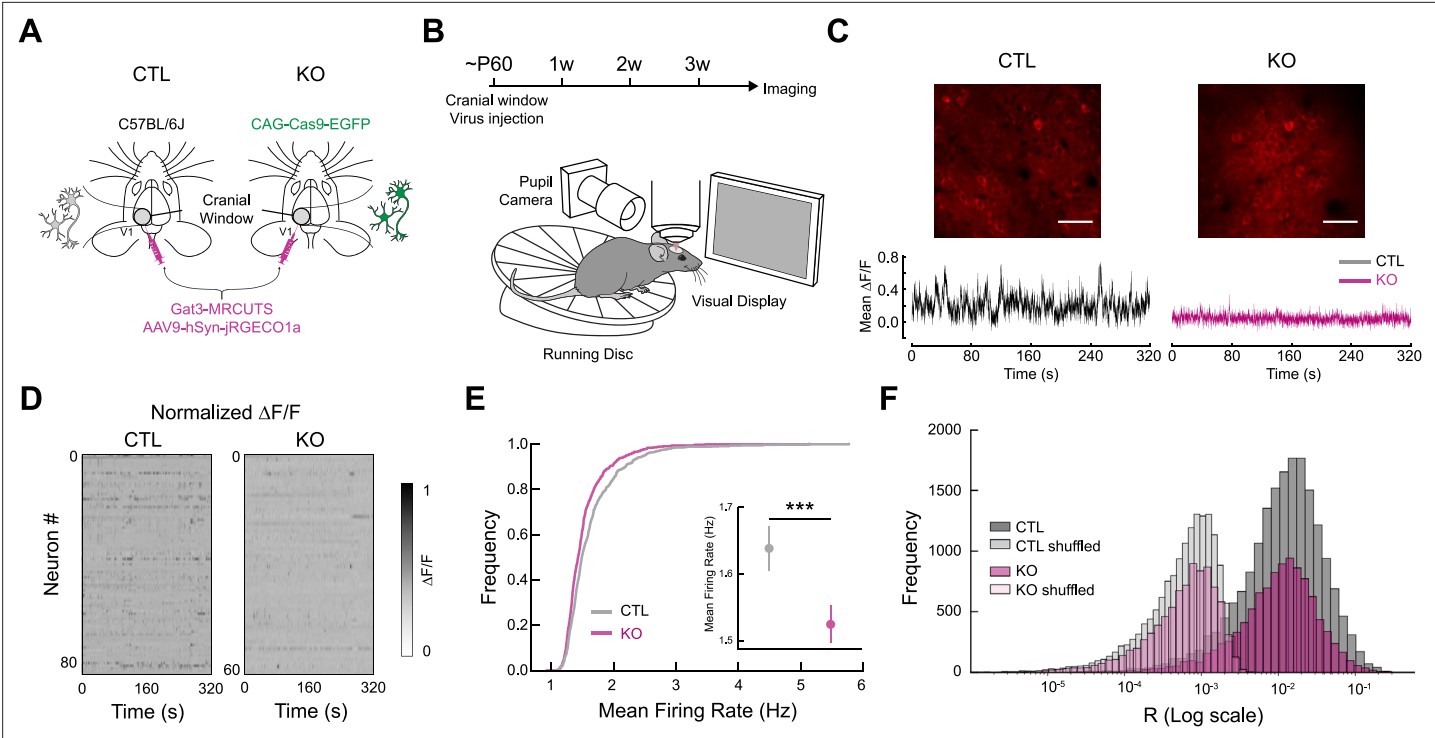

**Figure 3.** Genetic knockout of Gat3 in the visual cortex alters spontaneous activity of single neurons. (**A**) Schematic of control and experimental mice preparation. Both control mice (wild-type) and experimental mice (transgenic mice with cells constitutively expressing Cas9-EGFP under CAG promoter) received co-injection of Gat3-MRCUTS and a neuronal calcium sensor (jRGECO1a) in V1 during stereotactic surgeries. (**B**) Two-photon imaging setup consisting of a running wheel and a pupil camera to acquire locomotion and pupil dynamics, respectively. (**C**) Top: Example field-of-view (FOV) images of an imaging session from each group (scale bar = 50 μm). Bottom: Example average $Ca^{2+}$ traces from a control and a Gat3 KO mouse. The normalized $Ca^{2+}$ traces of all neurons within the same FOV were averaged. (**D**) Representative heatmaps of normalized spontaneous calcium activity of neurons in each session from each group. (**E**) Firing rates of individual neurons in control and Gat3 KO groups. Inset shows the average firing rates of all neurons from each group ($n_{control}$ = 838 neurons, 4 mice, $n_{Gat3 KO}$ = 606 neurons, 4 mice, ***p < 0.001, Linear mixed effects model (LME) $t$-stats, see Methods, error bars = SEM). (**F**) Distribution of pairwise correlation coefficients of neurons ($n_{control}$ = 31,460 pairs, $n_{Gat3 KO}$ = 18,004 pairs, n.s., p = 0.224, LME $t$-stats).

The online version of this article includes the following figure supplement(s) for figure 3:

**Figure supplement 1.** Comparisons of average firing rates of single neurons and neuron-to-neuron pairwise correlation during spontaneous activity.

## Region-specific knockout of Gat3 in the visual cortex alters response properties of individual neurons

Our *ex vivo* recordings showed an increased frequency of spontaneous inhibitory events in pyramidal neurons, suggesting enhanced inhibitory tone. However, how these changes at the cellular level impact neural representations and neural circuits in the intact, functioning brain remains unclear. To investigate the consequences of Gat3 ablation on neuronal dynamics *in vivo*, we performed two-photon calcium imaging of V1 neurons in awake mice, allowing us to monitor the activity of larger populations of neurons simultaneously while mice processed visual information. To achieve network-specific manipulation and to ensure that our astrocyte Gat3 KO was spatially limited to the neuronal population we were imaging, we co-injected Gat3-MRCUTS with a red-shifted calcium indicator (jRGECO1a) expressed under a neuronal promoter (hSyn) in V1 (*Figure 3A*). We also tracked locomotion and pupil dynamics to determine how Gat3 manipulation affects visual processing across dynamic brain states (*Figure 3B*).

We first determined whether Gat3 KO influenced spontaneous activity of neurons when mice were presented with a static gray screen. Neurons in the Gat3 KO animals had a noticeably reduced frequency of $Ca^{2+}$ transients, indicating lower spontaneous events (*Figure 3C*). Deconvolved calcium signals across all neurons in Gat3 KO animals showed a population-wide shift of the firing rate distribution toward lower firing frequencies (*Figure 3D, E*, *Figure 3—figure supplement 1A, B*). This reduced activity of predominantly excitatory neurons aligns with our *ex vivo* electrophysiological

findings showing increased spontaneous inhibitory inputs to pyramidal neurons (*Figure 2B–D*). Interestingly, despite these changes in individual neuronal firing rates, we found no significant alteration in neuronal synchrony or functional connectivity patterns, as measured by spontaneous pairwise correlation coefficients (*Figure 3F*, *Figure 3—figure supplement 1C*). The preservation of correlation structure suggests that Gat3 regulates the overall gain of neuronal activity without disrupting the underlying functional architecture of local circuits, pointing to a global modulation of excitability rather than a reorganization of specific connections.

Having established that Gat3 knockout alters spontaneous activity patterns in the visual cortex, we next investigated whether it also affects how neurons respond to visual stimuli. Visual encoding in the cortex requires precise coordination of excitation and inhibition (*Zhou and Yu, 2018*), and our findings thus far suggested Gat3 plays a role in regulating this balance. To systematically assess visual response properties, we examined neuronal activity in response to drifting gratings and natural movies, which allowed us to quantify critical aspects of visual encoding including response magnitude, orientation selectivity, and trial-to-trial reliability. Consistent with our observation of reduced spontaneous activity, neuronal responses to drifting gratings were attenuated in the Gat3 KO animals (*Figure 4A*). Furthermore, quantification of each neuron's maximum response to its preferred orientation revealed a significant reduction in peak response magnitude in Gat3 KO animals (*Figure 4B*, *Figure 4—figure supplement 1C*). Remarkably, despite this reduction in response magnitudes, the tuning properties of neurons remained broadly intact in Gat3 KO mice with no difference in tuning curves, average OSI, or percentage of highly selective cells (OSI ≥0.3) (*Scholl et al., 2013*; *Figure 4C–E*, *Figure 4—figure supplement 1A, B*). The dampened response magnitudes but not tuning suggest that Gat3-mediated regulation of ambient GABA modulates the response gain without disrupting underlying feature selectivity of cortical neurons.

The reliability of neural responses across repeated stimulus presentations is crucial for robust sensory processing; thus, we calculated reliability indices (*Rikhye and Sur, 2015*; see Methods) in order to evaluate whether Gat3 influences the trial-to-trial variability of individual neuronal responses to the same natural movie across multiple presentations (*Figure 4F, G*). Gat3 KO neurons exhibit greater trial-by-trial variability in the magnitude and timing of visual responses, as indicated by a significantly decreased reliability indices compared to control neurons (*Figure 4H*, *Figure 5—figure supplement 1A*). This finding reveals a critical role for astrocytic GABA transport in maintaining neuronal reliability over time.

Cortical neurons operate within a state-dependent framework where sensory responses are dynamically modulated by the animal's internal state (*Vinck et al., 2015*). V1 neurons not only encode visual stimuli, but also behavioral variables such as pupil-linked arousal and locomotion (*Vinck et al., 2015*; *Steinmetz et al., 2019*; *Engelhard et al., 2019*; *Osako et al., 2021*; *Stringer et al., 2019*). To test whether Gat3 ablation affects neuronal encoding of visual and non-visual information, we implemented a generalized linear model (GLM) (*Engelhard et al., 2019*; *Osako et al., 2021*; *Figure 4I*). While an increase in inhibitory tone on single neurons might intuitively suggest impaired information encoding, enhanced phasic inhibition can also sharpen neuronal response profiles and stimulus selectivity, potentially improving encoding performance (*Priebe, 2016*; *Shapley et al., 2003*; *Isaacson and Scanziani, 2011*; *Alitto and Dan, 2010*). This nuanced dynamic underscores why a GLM analysis is necessary to directly measure the outcome of these single neuronal changes.

We constructed GLMs to simultaneously quantify the relative contributions of multiple factors/predictors (visual stimuli, pupil diameter, and locomotion) to neuronal activity through the model's explained variance ($R^2$). By comparing the explained variance ($R^2$) of the full model and individual predictors (see Methods) in neurons after Gat3 KO, we can determine if Gat3 selectively impacts particular information channels, or if it impacts encoding more globally and uniformly. When considering explained variance ($R^2$) of the full model with all predictors, we found that both the distribution of $R^2$ values (*Figure 4J*) and the mean $R^2$ across neurons (*Figure 4K*, *Figure 4—figure supplement 1D*) were significantly decreased in Gat3 KO animals compared to controls, suggesting a general impairment in encoding across these major predictors. To determine whether Gat3 ablation differentially affected the encoding of specific types of information across the neuronal population, we calculated the proportion of neurons that significantly encoded each individual predictor. While we observed a slight decrease in the proportion of cells encoding visual stimuli and locomotion in the Gat3 KO group, these differences did not reach statistical significance (*Figure 4L*), suggesting that

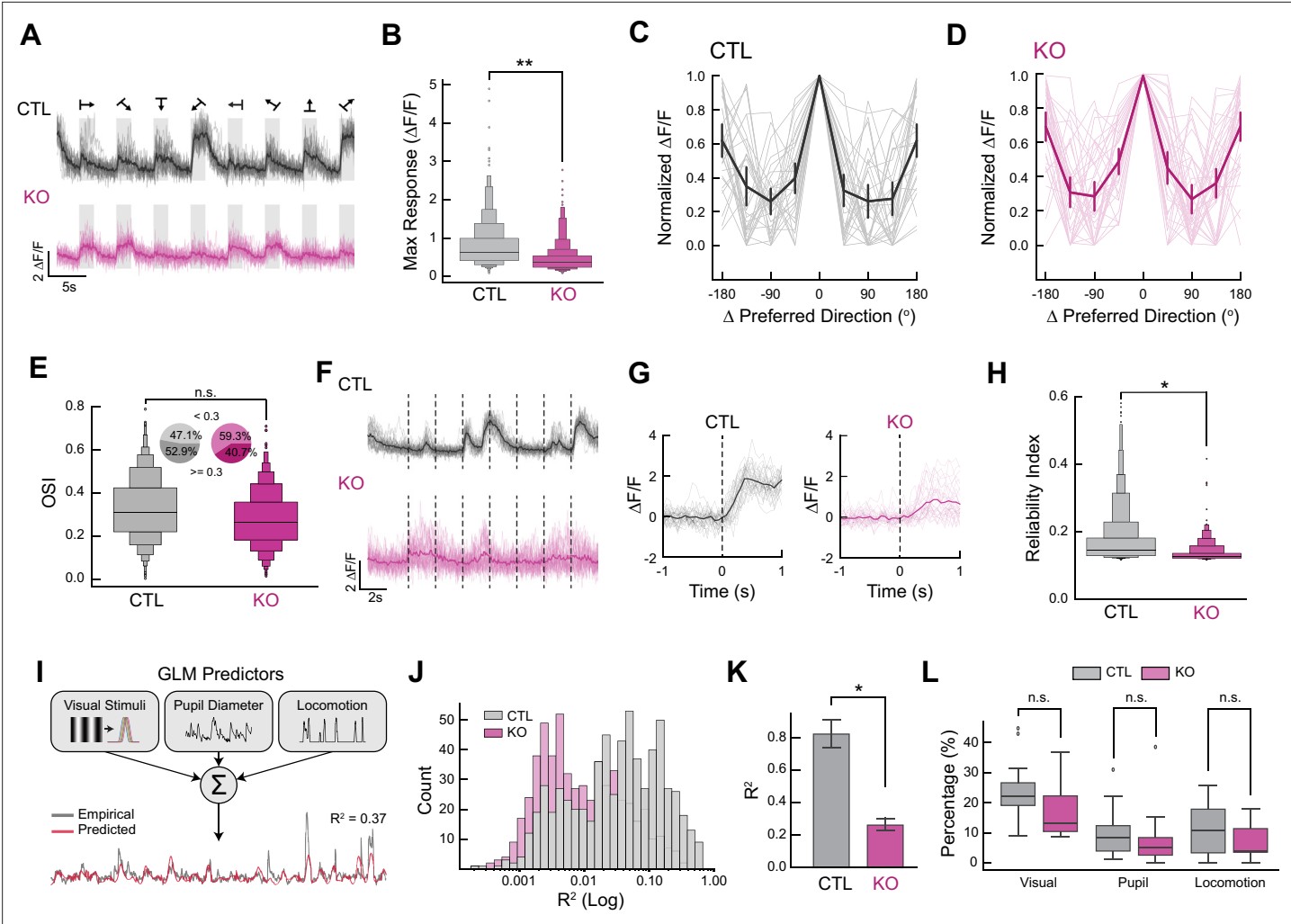

**Figure 4.** Genetic knockout of Gat3 in the visual cortex alters the visual response properties of neurons. (**A**) Example Ca²⁺ traces of a single neuron from control (top) and Gat3 KO (bottom) during presentation of drifting gratings. The average of all trials is plotted in a dark line overlaid on the lighter individual trial traces (16 trials in total). (**B**) Average maximum response magnitudes of neurons to their preferred grating orientation. Visually responsive neurons were pooled across animals within each group ($n_{control}$ = 526 neurons, 4 mice, $n_{Gat3\ KO}$ = 366 neurons, 4 mice, **p < 0.01, LME $t$-stats). (**C**) Representative tuning curves of control individual neurons (in lighter shade) and the average tuning curve (in bold) of all neurons in each FOV centered around their preferred orientation ($n_{control}$ = 32 neurons, error bars = SEM). (**D**) Same as B but for Gat3 KO ($n_{Gat3\ KO}$ = 38 neurons, error bars = SEM). (**E**) Comparison of orientation selectivity index (OSI) distribution of visually responsive neurons between the two groups (n.s., p = 0.183, LME $t$-stats). Insets show the percentage of cells with OSI greater or less than 0.3. (**F**) Example Ca²⁺ traces of a single neuron from control (top) and Gat3 KO (bottom) during natural movies where the dotted lines indicate the onset of a movie. The average of all trials is plotted in a dark line overlaid on the lighter individual trial traces (32 trials in total). (**G**) Example plots showing variability of each trial response (in lighter shade) of a single neuron to a natural video; dotted line indicates the stimulus onset. (**H**) Reliability indices of neurons to their preferred stimuli in control and Gat3 KO group ($n_{control}$ = 707 neurons, 4 mice, $n_{Gat3\ KO}$ = 436 neurons, 4 mice, *p < 0.05, LME $t$-stats). (**I**) Generalized linear model (GLM)-based single neuron encoding model of visual stimulus information, pupil dynamics, and running speed. Variance explained ($R^2$) is computed to assess the encoding property of neurons. (**J**) Distribution of $R^2$ of individual neurons from each group ($n_{control}$ = 647 neurons, 4 mice, $n_{Gat3\ KO}$ = 565 neurons, 4 mice). (**K**) Comparison of average $R^2$ values of individual neurons between the two groups (*p < 0.05, LME $t$-stats, error bars = SEM). (**L**) Proportions of neurons encoding each parameter (visual stimuli, pupil dynamics, and movement) from each imaged population (n.s., $p_{Visual\ stimuli}$ = 0.116; n.s., $p_{Pupil}$ = 0.662; n.s., $p_{Movement}$ = 0.172, LME $t$-stats).

The online version of this article includes the following figure supplement(s) for figure 4:

**Figure supplement 1.** Comparisons of visual responses of single neurons to drifting gratings.

Gat3 reduction affected neural encoding more broadly rather than impacting specific information channels. This general reduction in encoding capacity aligns with our earlier observation of increased trial-to-trial variability (*Figure 4H*), while extending the scope of reduced neural encoding to continuously varying readouts of internal states such as pupil diameter and locomotion. Together, these results indicate that Gat3 is important for maintaining reliable neuronal responses regardless of the information being encoded.

## Gat3 reduction impairs information encoding of neuronal populations in visual cortex

Given the complexity of microcircuits and heterogeneity of cell types in the cortex, Gat3 manipulation may exert changes in population dynamics that are not captured by our single neuron-level analyses. To investigate these potential network effects, we first examined how functional interactions between pairs of neurons might be influenced by Gat3 knockout. Specifically, we computed two correlation metrics in response to natural movies: signal correlation, which quantifies tuning similarity between neurons, and noise correlation, which measures co-fluctuation of trial-to-trial variability (*Figure 5—figure supplement 1B*, see Methods; *Cohen and Kohn, 2011*; *Panzeri et al., 2022*).

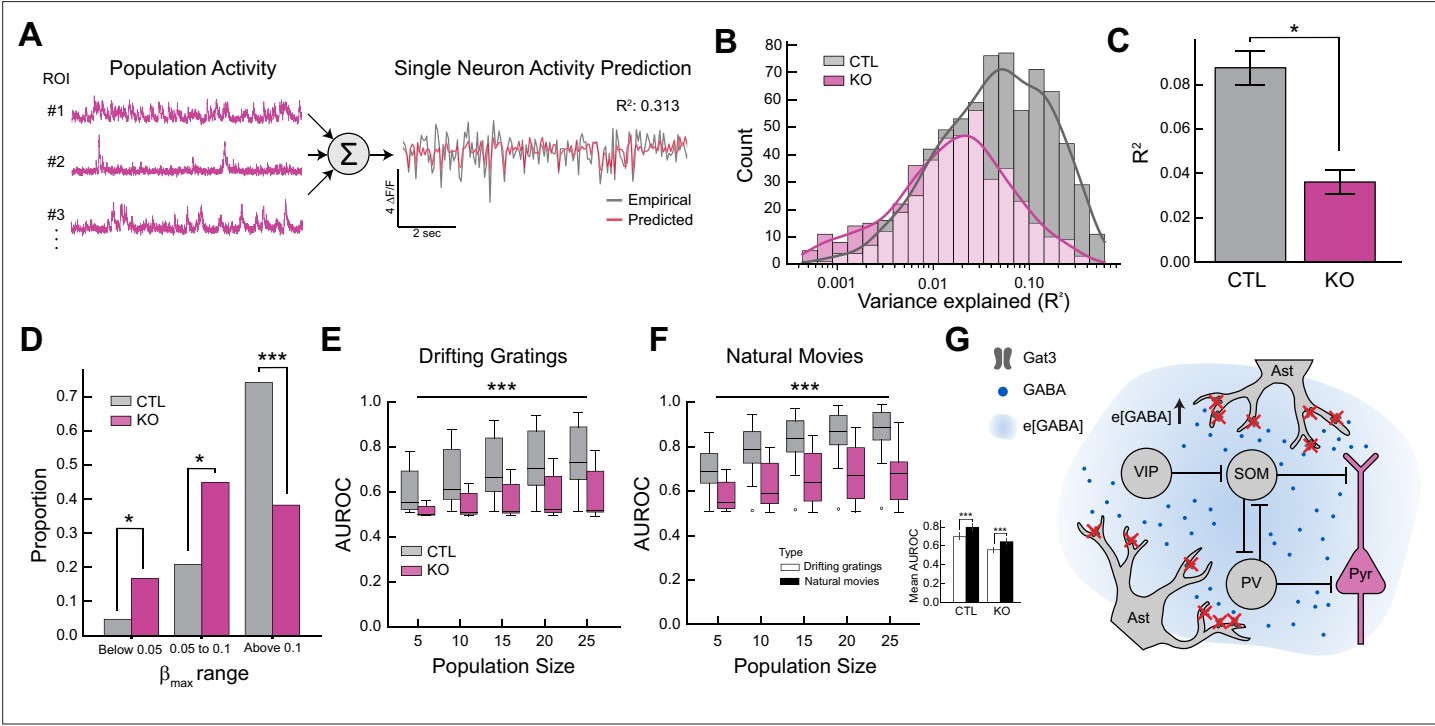

**Figure 5.** Genetic knockout of Gat3 alters population-level properties of cortical neurons. (**A**) Schematic of a single neuron encoding model of population activity using generalized linear model (GLM). Calcium traces of randomly sampled neurons in a fixed population size were used to train a GLM model for prediction of the target neuron's activity. (**B**) Distribution of $R^2$ values of individual neurons ($n_{control}$ = 707 neurons, 4 mice, $n_{Gat3\ KO}$ = 436 neurons, 4 mice, training population size = 20 neurons). (**C**) Comparison of average $R^2$ value of all neurons between two groups (*p < 0.05, LME *t*-stats, error bars = SEM). (**D**) The maximum value of the predictor weights (b) from each neuron's GLM fitting was extracted and grouped into ranges of below 0.05, 0.05–0.1, and above 0.1. The difference in proportions of the weights showed the different level of encoding of other neurons between the two groups (*p < 0.05, ***p < 0.001, Mann–Whitney *U* test). (**E**) Support Vector Machine (SVM)-based decoding analysis of neuronal population activity induced by drifting gratings in neuronal populations of various sizes. Comparison of decoding accuracy of visual stimulus information (area under the receiver operating characteristic curve) of populations between two groups ($n_{control}$ = 12 sessions, 4 mice, $n_{Gat3\ KO}$ = 9 sessions, 4 mice, ***p < 0.001, two-way ANOVA). (**F**) Same as E but for natural movies ($n_{control}$ = 11 sessions, 4 mice, $n_{Gat3\ KO}$ = 11 sessions, 4 mice, ***p < 0.001, two-way ANOVA). Inset: comparison of average area under the receiver operator characteristic curve (AUROC) between different visual stimuli within each group (***p < 0.001, Mann–Whitney *U* test, error bars = SEM). (**G**) A simplified diagram of a visual cortex L2/3 microcircuit consisting of neurons and astrocytes. The microcircuit contains different types of inhibitory neurons that exert inhibitory or disinhibitory effects on pyramidal neurons. Extra-synaptic expression of Gat3 in astrocytic processes allows astrocytes to control extracellular GABA levels that may differentially influence a wide network of cells.

The online version of this article includes the following figure supplement(s) for figure 5:

**Figure supplement 1.** Comparisons of neuronal responses to natural movies.

Signal correlations of Gat3 KO neurons did not significantly differ from those of control neurons, suggesting that the tuning similarity of different neurons to the same set of stimuli was not affected by Gat3 ablation (*Figure 5—figure supplement 1C*). This preservation of stimulus-dependent correlations across neuronal pairs indicates that the microcircuit supporting feature selectivity across the population persists even when ambient GABA levels may be altered by loss of Gat3. Similarly, noise correlations, reflecting trial-to-trial variability in neuronal responses, of the Gat3 KO neurons were not significantly different from those of the control neurons, although there was a general trend of reduced correlation coefficients in the KO group (*Figure 5—figure supplement 1D*). These results indicated that Gat3 does not significantly influence the functional connectivity between pairs of neurons.

While the pairwise correlations across the entire population were not significantly altered, correlations between clusters of neurons could still be influenced by the loss of Gat3. To evaluate how individual neurons may be influenced by the activity of other neurons in the population, we used a GLM to predict a target neuron's activity using neural activity from other neurons in the population during the presentation of natural movies (*Figure 5A*). Unlike pairwise correlation analysis, a GLM can identify which specific neurons best predict a target neuron's activity and adjust their weights accordingly, allowing the identification of more correlated neuronal clusters. To ensure robust comparison between control and Gat3 KO conditions, we systematically varied the number of predictor neurons used in the model, assessing how the encoding performance of a single-neuron scales with input population sizes. We observed that Gat3 KO neurons have a lower $R^2$, suggesting that activity from other neurons in the population serves as a poor predictor for neural activity (*Figure 5B, C*, *Figure 5—figure supplement 1E, F*). The proportion of neurons with strong predictive connections (weights >0.1) was also significantly lower in the Gat3 KO group (*Figure 5D*). This finding suggests that in Gat3 KO mice, individual neurons operate more independently with less coordinated activity with neighboring neurons. This likely reflects greater overall noise and disruption of functional clusters that normally process information collectively. Thus, Gat3 appears to regulate network-wide coordination of neuronal activity rather than modulating specific neuronal connections.

Our analyses of visual encoding thus far have focused on the contributions of single neurons but have yet to address how visual information may be represented across the entire population. While we observed no change in the distribution of individual neuron OSI with Gat3 KO (*Figure 4E*), the reduced neuronal responsiveness (*Figure 4A, B*) and population correlations (*Figure 5C*) caused by Gat3 ablation can influence population-level encoding of visual information. To examine how Gat3 influenced visual representations at the population level, we used Support Vector Machine (SVM) classifiers to decode stimulus identity from patterns of neuronal activity (*Scott et al., 2017*). In the control group, decoding accuracy of the visual stimulus progressively improved with increasing population size of predictors, indicating that larger populations of neurons provide a more robust representation of visual information (*Figure 5E*). In contrast, the neuronal populations in the Gat3 KO group demonstrated consistently low performance, with the mean area under the receiver operating characteristic curve (AUROC) remaining only marginally above chance, even with an increased number of neurons in the population. This suggests that the observed reduced correlations (*Figure 5C*, *Figure 5—figure supplement 1E, F*) were not due to unique stimulus features carried by individual neurons, as would be expected if decorrelation served to reduce redundancy and enhance population coding efficiency, but instead reflected degraded coordinated activity patterns necessary for robust stimulus decoding across the population. This trend was also observed in the decoding of natural movies, with Gat3 KO significantly reducing decoding performance (*Figure 5F*). Nevertheless, we note that the decoding performance for natural movies exceeded that for drifting gratings in both control and KO groups (*Figure 5F*, inset). This is likely because the complex visual components in natural movies require dynamic coordination across neuronal populations, while strongly oriented edges represented by drifting gratings are well encoded by single neurons (*Graf et al., 2011*; *Tanabe, 2013*; *Montijn et al., 2016*). The decoding deficits following Gat3 ablation provide evidence that astrocytic regulation of ambient GABA is essential for organizing the coordinated neuronal activity patterns necessary for efficient information encoding in visual cortical networks.

## Discussion

In this study, we provide the first *in vivo* investigation of Gat3 function across multiple levels of neuronal activity and representations. Using our newly developed multiplexed CRISPR technique

(MRCUTS), we knocked out Gat3 expression in the visual cortex to explore its role within intact neural circuits in awake mice processing visual stimuli (*Figure 2*). Our strategy allows us to achieve Gat3 KO with genetic and spatial specificity while avoiding off-target effects commonly associated with pharmacological approaches. We found that Gat3 knockout increased spontaneous inhibitory postsynaptic currents in pyramidal neurons (*Figure 2*), aligned with reduced calcium transients *in vivo* (*Figure 3*), and diminished both magnitude and reliability of responses to visual stimuli (*Figures 3 and 4*). These changes at the single neuron level extended to the network level, where Gat3 knockout impaired the information encoding capacity of neuronal populations in the visual cortex (*Figure 5*). These effects on cortical microcircuits (*Figure 5G*) influenced the robustness of population encoding, as evidenced by our population analyses. Given that Gat3 expression is limited to astrocytes (*Melone et al., 2014*; *Scimemi, 2014*; *Minelli et al., 1996*), these findings reveal a critical role for astrocytic Gat3 in maintaining the reliability of sensory representations at both single neuron and population levels. They highlight that astrocytes act not merely as passive supporters of neural function but as active regulators of information encoding in cortical circuits (*Cahill et al., 2024*; *Nagai et al., 2021*; *Kofuji and Araque, 2021*). By modulating the inhibitory tone across neuronal populations such as with Gat3, astrocytes help establish the optimal excitation–inhibition balance necessary for reliable sensory processing. Future work should explore neuronal activity in more defined cell types to resolve the microcircuit implications of impaired tonic inhibition as different cell types might have differential sensitivity to changes in ambient GABA. Such studies may reveal how tonic inhibition may shift the balance between different interneuron classes to alter network dynamics and stimulus encoding.

Previous studies support our findings by demonstrating that Gat3 modulates both excitatory and inhibitory neuronal properties, primarily via changes in tonic and phasic GABA conductance and neuronal excitability (*Kersanté et al., 2013*; *Song et al., 2013*; *Boddum et al., 2016*; *Chazalon et al., 2018*). While a direct measurement of tonic GABAergic current may have revealed the changes in ambient GABA level, we have prioritized sIPSC measurements as tonic GABA currents have been shown to strongly correlate with phasic inhibitory bursts (*Glykys and Mody, 2007*; *Farrant and Nusser, 2005*; *Ataka and Gu, 2006*) and sIPSCs can reflect a more integrative and reliable proxy for altered GABAergic signaling. In line with the earlier observations (*Song et al., 2013*; *Shigetomi et al., 2012*; *Chazalon et al., 2018*; *Kinney, 2005*), our *ex vivo* patch clamp recording experiments showed that Gat3 reduction increases sIPSC frequency in cortical pyramidal neurons, possibly due to (1) increased ambient GABA diffusing into the synaptic cleft; (2) desensitization of presynaptic $GABA_B$ receptors that normally have inhibitory effects on GABA release; and (3) decreased disinhibition, which may increase the inhibitory output onto excitatory neurons (*Kinney, 2005*). These cellular effects align with our *in vivo* calcium imaging results showing decreased neuronal activity in Gat3 KO. Astrocytes in the hippocampus can shift network excitation to tonic inhibition via Gat3 reversal (*Héja et al., 2012*) and altered Gat3 expression in the striatum can modify medium spiny neuron activity underlying behavioral phenotypes (*Yu et al., 2018*). While these findings are consistent with our results, most prior studies focused on probing Gat3 function in subcortical regions using pharmacological manipulations.

We found through our population decoding analyses that Gat3 KO degraded population coding despite maintaining single neuron stimulus tuning (OSI), revealing an intriguing dissociation between preserved feature selectivity and impaired information encoding. This suggests that population sensory representations require not only appropriately tuned neurons but also coordinated activity across the population. Furthermore, Gat3 KO had a more global effect in degrading neural encoding, rather than specifically affecting visual encoding or movement modulation, underscoring that astrocytic regulation of inhibitory transmission pervades multiple domains of the cortical circuit. Notably, despite degraded population coding, pairwise correlations appeared preserved upon Gat3 KO. This indicated that standard pairwise correlation metrics may not capture higher-order patterns of neuronal coordination across clusters of neurons disrupted by Gat3 KO (*Averbeck et al., 2006*). Our findings thus point to a more sophisticated role for astrocytic GABA transport in maintaining the complex temporal organization of neuronal ensemble activity that extends beyond what can be captured by standard correlation analyses. This is consistent with the notion that tonic inhibition can modulate the temporal precision of spike timing and the synchronization of neuronal assemblies without necessarily affecting the overall correlation structure.

Our study revealed that localized Gat3 ablation disrupted cortical dynamics. However, we acknowledge potential secondary effects despite the specificity of our approach. A number of studies, including our own, have shown that Gat3 expression is not static (*Cho et al., 2022*)—it varies across development and as a result of astrocytic $Ca^{2+}$ activity (*Shigetomi et al., 2012*; *Yu et al., 2018*; *Kang et al., 2023*). These findings raise important questions: Are the observed changes in Gat3 expression driven by activity-dependent mechanisms, homeostatic processes, or a combination of the two? Furthermore, it remains unclear how genetic disruption of Gat3 might influence interactions with other key proteins. The alterations in neuronal population dynamics could result not only from the direct loss of Gat3 but also from secondary effects involving related molecules such as receptors, channels, or transporters (*Song et al., 2013*; *Scimemi, 2014*; *Shigetomi et al., 2012*). We cannot rule out compensatory mechanisms, including potential upregulation of other GABA transporters such as Gat1. These factors underscore the need for further investigation to unravel the specific pathways through which Gat3 reduction affects both neuronal and broader network functions.

Our study introduces MRCUTS, the first genetic tool utilizing multiplexed CRISPR design for astrocyte manipulation, which is compatible with commercially available transgenic mouse models. In general, astrocytes have received less attention in CRISPR/Cas9 tool development compared to neurons; thus, MRCUTS provides a unique method to manipulate astrocytic genes *in vivo* simultaneously. Here, we leveraged Gat3's exclusive expression in astrocytes of adult mouse brain (*Melone et al., 2014*; *Scimemi, 2014*; *Minelli et al., 1996*; *Pow et al., 2005*) to use constitutive Cas9-expressing mice rather than requiring astrocyte-restricted Cas9 expression. This design choice provides several advantages over tamoxifen-inducible Cre-dependent Cas9 systems: more consistent expression levels across cells, elimination of tamoxifen's potential confounding effects on neurophysiology, and a simplified experimental timeline that avoids variable recombination efficiency and delay period associated with inducible systems. This approach also allowed us to avoid potential limitations of using astrocyte-specific promoters, which can vary in their expression patterns or be influenced by disease states and inflammatory responses. However, we highlight that our MRCUTS approach maintains versatility for use in various Cas9-expression systems to achieve cell-type specific knockout of genes expressed in multiple cell types. Although this study focused solely on Gat3, MRCUTS can be developed to target other genes or gene combinations, offering a powerful tool for investigating astrocytes in brain function and behavior.

## Materials and methods

### Animals

All experimental procedures performed in this study were approved by the Massachusetts Institute of Technology's Animal Care and Use Committee (protocol #2308-000-562) and conformed to the National Institutes of Health. Adult mice (2–4 months) were housed on a reverse 12 hr light/dark cycle with controlled temperature and ventilation. C57BL/6JA wild-type mice (Stock No. 000664, Jackson Laboratory) were used as control and CAG-Cas9-EGFP (B6J.129(B6N)-Gt(ROSA)26Sortm1(CAG-cas9*-EGFP)Fezh/J) (Stock No. 026175, Jackson Laboratory) were used for constitutive expression of Cas9 protein in all cells.

### Multiplex CRISPR KO construct design and AAV vector generation

CRISPR/Cas9 knockout (KO) sgRNAs were designed to target multiple obligatory protein coding exons of the mouse *Gat3* (*Slc6a11*, *ENSMUST00000032451.9*) gene to create insertion/deletions that abolish the gene's protein output. Two KO sgRNAs were placed adjacently within each exon to maximize gene deletion efficiency. The KO sgRNA pairs were placed on exons 1, 2, and 5 of the mouse *Gat3* gene based on mRNA transcript NM172890. For each Gat3 KO sgRNA sequence, a Csy4 enzyme target site sequence GTTCACTGCCGTATAGGCAG was added to its 5′ end, and a universal CRISPR/Cas9 sgRNA backbone sequence to its 3′ end. A U6::6X Gat3 KO sgRNAs cassette was designed to include a single U6 promoter, followed by six Csy4-sgRNA-backbone sequences linked in tandem. The gene fragment was synthesized by GenScript. A PX552 AAV vector backbone (Addgene #60958) was modified to insert a PGK::His tag-Csy4 expression cassette upstream of the hGH PolyA 3′ UTR to replace the U6::sgRNA scaffold-hSyn::EGFP segment of the original plasmid. A number of restriction enzyme sites were placed at the 5′ end of the PGK promoter for inserting the U6::6X Gat3 KO

sgRNAs cassette through T4 ligation. We term the multiplex CRISPR KO construct: Multiple sgRNA Csy4-mediated Universal Targeting System (MRCUTS). AAV virus particles were packaged using the service of UNC viral core facility (AAV8.2, titer = 2E + 13 vg/ml). We refer to the virus containing the multiplexed CRISPR construct targeting Gat3 as Gat3-MRCUTS in the following sections.

Mouse *Gat3* gene KO sgRNA sequences were as follows:

CGGCCACTGGAACAACAAGG, AAAACACCACGTAAGGAATC, ATAATGCCAGTTCCCAACGG, TCATCGGACTGGGCAACGTG, TTCTTCCTGGAAACGGCTCT, TGGAAGGGTACTAAGTCGAC.

## Glial cell culture and Western blot

Primary astrocyte cultures were prepared from 1 or 2-day-old neonatal C57BL/6 mice of both sexes. Brains were removed after decapitation and cortices were dissected out using a dissecting microscope. The tissues were dissociated in the enzymatic solution containing papain for 30 min at 37°C followed by mechanical trituration to obtain a single cell suspension. The collected cell suspension was cultured in 6-well plates pre-coated with 50 μg/ml poly-D-lysine in Astrocyte medium (ScienCell) supplemented with 10% fetal bovine serum and was maintained at 37°C in a humidified atmosphere containing 5% $CO_2$. The medium was changed every 3–4 days. To examine the knockout efficiency of our multiplexed CRISPR construct, astrocytes were transfected with PX458 spCas9-P2A-GFP and PX552 PGK-Csy4 6XsgRNA constructs at 10 days *in vitro* using Lipofectamine 3000 (Thermo Fisher Scientific) following manufacturer's protocol. Briefly, for transfection in a single well of a 6-well plate, 3 μg of DNA was mixed with 6 μl of Lipofectamine 3000 in 250 μl opti-MEM (Thermo Fisher Scientific).

Western blot assays were performed 5 days after transfection. Cells were washed with ice-cold phosphate-buffered saline (PBS) and lysed in 1X RIPA lysis buffer (Abcam) containing protease inhibitor cocktail (Sigma). Samples were subsequently boiled in 1X NuPAGE LDS sample buffer (Thermo Fisher Scientific) at 99°C for 10 min and proteins were separated on NuPAGE 4–12% Bis-Tris Gels (Thermo Fisher Scientific). They were then transferred onto a PVDF membrane using the iBlot system (Thermo Fisher Scientific) and blocked with 5% nonfat dry milk in 0.05% PBS-T (PBS-Tween 20) for 1 hr at room temperature. Immunoblotting was then performed by incubating the membrane with either anti-GAT3 rabbit polyclonal antibody (1:1000, Synaptic Systems) or anti-β-actin mouse monoclonal antibody (1:1000, Sigma) at 4 °C overnight. Membranes were washed and incubated with the HRP-conjugated secondary antibodies (1:5000, Cell Signaling Technology) for 45 min at room temperature. Detection was carried out using the Immobilon Forte Western HRP substrate (MilliporeSigma) and images were acquired with ChemiDoc Imaging system (Bio-Rad). Data was quantified by densitometric analysis using ImageJ software and individual band intensities were normalized to β-actin.

## Stereotactic surgeries

Surgeries were performed under isoflurane anesthesia (3% for induction, 1–1.5% for maintenance) while maintaining body temperature at 37.5°C. Mice were given pre-emptive slow-release buprenorphine (1 mg/kg, s.c.) and post-operative meloxicam (5 mg/kg, s.c.). Mice were head-fixed in a stereotaxic frame, scalp hair was removed, and the skin was sterilized with 70% ethanol and betadine. A portion of the scalp was removed to expose the skull.

For virus injections, we drilled 3 small craniotomies (0.5 mm) per hemisphere centered around the coordinates for the visual cortex (1.5 mm anterior and 2.5 mm lateral to lambda) and injected a volume of 200–300 nl per site at a rate of 60 nl/min 0.2 mm below the pial surface with a glass pipette and a stereotaxic injector (QSI 53311, Stoelting). The glass pipette was left in place for an additional 5 min after the injection and was slowly withdrawn to avoid virus backflow. Injections were performed in both hemispheres to maximize tissue collection. Gat3-MRCUTS and AAV1-CAG-tdTomato were co-injected to visualize the injection site for tissue collection.

For two-photon imaging experiments, a round 3 mm diameter craniotomy was performed over the left visual cortex (1.5 mm anterior and 2.5 mm lateral to lambda) and a total volume of 300–400 nl of virus, split over 3 injection sites, at 60 nl/min was injected 0.2 mm below the pial surface. Gat3-MRCUTS and AAV9-hSYN-jRGECO1a were co-injected for neuronal expression of red calcium indicator. A cranial window was prepared with 3 round coverslips (CS-5R, 1 × 5 mm diameter; CS-3R, 2 × 3 mm diameter; Warner Instruments) glued together with UV-cured adhesive (catalog #NOA 61, Norland). The cranial window was implanted over the 3 mm craniotomy and sealed with dental cement (C&B Metabond, Parkell). A headplate was fixed to the skull using the same dental cement for

calcium imaging. Mice were monitored for 3 days following surgery. Mice recovered for at least 5 days before habituation and imaging experiments. After completion of experiments, mice were perfused for *post hoc* validation of Gat3 knockout efficiency.

## DNA sequencing

Mice were deeply anesthetized under isoflurane and decapitated for rapid brain extraction. Cortical regions labeled with a fluorescent marker were dissected in ice-cold 0.9% saline and meninges were removed. Cortex biopsies were flash frozen and stored at −80°C. Frozen samples were later equilibrated to room temperature, cut into small pieces (≤25 mg), and genomic DNA was extracted using QIAamp DNA Mini Kit (QIAGEN) according to the manufacturer's instructions. Genomic sites of interest were PCR amplified from purified genomic DNA using Q5 High-Fidelity DNA Polymerase (NEB) with the following primers flanking the gRNA-targeting regions of Gat3: GAT3_Locus1 Forward, GCCA TGACTGCGGAGCAAGC, Reverse, ATGCACGAGAGGTGTCACCCCAC; GAT3_Locus2 Forward, TGGAATTCCAGCTGAAAGAGGGCCGT, Reverse, TCCTTTGAAACAGCCTTGGCAGCT. Amplicons were then sequenced using Primordium Premium PCR sequencing. Reads per base data was used to quantify the number of variants in the gRNA-targeting areas from the same pool of PCR amplicons within each sample, and results from one gRNA-targeted region in Gat3 KO mice were represented as normalized to control.

## Immunohistochemistry

Mice were transcardially perfused with 0.9% NaCl followed by 4% paraformaldehyde. Coronal sections were sectioned with a vibratome at 50 mm thickness. Brain sections were incubated in a blocking solution (1% Triton X-100, 5% BSA in PBS) for 1 hr at room temperature on a shaker. Sections were incubated in the blocking solution containing primary antibody rabbit anti-GAT3 (1:500, catalog #274 304, SySy) overnight at 4°C. Sections were washed and incubated for 4 hr in blocking solution with secondary antibody goat anti-rabbit Alexa Fluor 647 (1:1000, catalog #A32733, Invitrogen). Sections were washed in PBS and mounted on slides in DAPI-containing mounting medium (VECTASHIELD, catalog #H-1500, Vector Laboratories). Images were taken using a Leica confocal microscope (TCS SP8, Leica) with a 20x/0.75 NA objective lens and LAS X Acquisition Software (Leica). The images were processed with the ImageJ software.

## Slice preparation

Adult mice (WT or CAG-Cas9-EGFP) were injected with Gat3-MRCUTS targeting layer 2/3 of the visual cortex (AP: 1.5 mm; ML: 2.5 mm; DV: 0.2 mm from lambda). Mice were anesthetized using isoflurane, and the brains were rapidly dissected out and transferred to oxygenated (95% $O_2$/5% $CO_2$), ice-cold cutting solution containing (in mM): 118 Choline chloride, 2.5 KCl, 1.2 $NaH_2PO_4$, 26 $NaHCO_3$, 10 glucose, 2.5 $CaCl_2$, and 1.3 $MgCl_2$. Coronal slices (300 μm thick) containing the visual cortex were cut using a Leica VT1200S vibrating blade microtome, transferred to an oxygenated solution containing (in mM): 92 NaCl, 2.5 KCl, 1.2 $NaH_2PO_4$, 30 $NaHCO_3$, 20 HEPES, 25 glucose, 2 Thiourea, 5 Na-ascorbate, 3 Na-pyruvate, 5 *N*-acetyl-L-cysteine, 2 $CaCl_2$, and 2 $MgCl_2$ and allowed to recover for 1 hr. For electrophysiological recordings, slices were transferred to a superfused recording chamber, constantly perfused with oxygenated aCSF containing (in mM): 115 NaCl, 10 glucose, 25.5 $NaHCO_3$, 1.05 $NaH_2PO_4$, 3.3 KCl, 2 $CaCl_2$, and 1 $MgCl_2$ and maintained at 28°C.

## Whole-cell patch recordings

Whole-cell voltage-clamp recordings were performed on neurons in layer 2/3 of the visual cortex with pipettes (3–5 MΩ resistance) pulled from thin-walled Borosilicate glass using a Sutter P97 Flaming/ Brown micropipette puller. The pipettes were filled with an internal solution containing (in mM): 140 Cesium chloride, 10 HEPES, 0.5 EGTA, 2 $MgCl_2$, 10 phosphocreatine, 5 Mg-ATP, and 1 Na-GTP. The pH of the internal solution was adjusted to 7.3 with Cesium hydroxide and osmolarity adjusted to 295–300 mOsm. All recordings were obtained using a MultiClamp 700B (Molecular Devices) amplifier and digitized using the Digidata 1440A system (Molecular Devices). Signals were filtered at 2 kHz and digitized at 10 kHz. Neurons were included in the study only if the initial resting membrane potential (Vm) was ≤ −55 mV, access resistance (Ra) was <25 MΩ and were rejected if the Ra changed by >20% of its initial value. For all recordings, neurons were held at −65 mV. sIPSCs were isolated by

blocking excitatory currents with 10 µM 6-Cyano-7-nitroquinoxaline-2,3-dione disodium salt hydrate (CNQX) and 30 µM D(−)-2-Amino-5-phosphonopentanoic acid (D-AP5) perfused in the bath. Continuous current traces of 5 min duration (recorded at least 5 min after achieving whole-cell configuration) were acquired and analyzed with Clampfit 10.7 (Molecular Devices). The frequency and amplitude of recorded sIPSCs were quantified.

### *In vivo* two-photon imaging

Mice were head-fixed in a behavior rig which consisted of a running wheel and a screen monitor on the right side of the wheel for presentation of visual stimuli and imaging of the left visual cortex. Imaging was performed 3–5 weeks post-virus injection and after 2–3 days of habituation. A Prairie Ultima IV two-photon microscopy system was used with a resonant-galvo scanning module (Bruker). Two-photon excitation of jRGECO1a at a 1020-nm wavelength was provided by a tunable Ti:Sapphire laser (Mai Tai eHP, SpectraPhysics) and the signal was collected by GaAsP photomultiplier tubes (Hamamatsu). Images were acquired with a 16x/0.8 NA microscope objective (Nikon) with 2x optical zoom at 16 Hz. A 512 × 256 pixel FOV was imaged for each imaging session of awake mice. The mice were presented with 3 sets of visual stimuli, a gray screen, drifting gratings, and natural movies, in 3 different imaging sessions. Movement of the mice on the wheel was tracked and recorded simultaneously. A pupil camera was placed next to the screen monitor to record pupil dynamics during the sessions.

### Visual stimuli

Isoluminant visual stimuli (static gray screen, drifting bar gratings, and natural movies) were constructed by MATLAB (Mathworks) using Psychtoolbox. For spontaneous activity, a gray screen was shown for 640 s. For drifting gratings, a set of drifting gratings was presented for 2 s following a 3 s OFF period, with gratings rotated by 45° for a set of four orientations and eight directions. A single trial consisted of a set of 8 grating directions, each separated by an OFF period, with 16 trials in one session. For natural movies, we used a total of seven different black-and-white movies from the van Hateren movie database (*Rikhye and Sur, 2015*; *van Hateren and Ruderman, 1998*). Each trial started with a single OFF period for 3 s and was followed by a consecutive presentation of individual movies, each being presented for 2 s. The order of the movies remained the same, and the trials were repeated 32 times in a session.

### Calcium data analysis

After image acquisition, time-lapse imaging sequences were corrected for motions using a template-matching ImageJ plug-in. We used a two-photon calcium imaging analysis pipeline, Suite2P (*Pachitariu et al., 2017*), to detect cells in the recorded data. Detected ROIs were manually curated to select ROIs with putative somatic activity during the imaging session. Preprocessing and basic visual response property analysis of the data (i.e. orientation selectivity and tuning curves) were carried out using custom functions written in MATLAB. We acquired the somatic fluorescence signal from the ROI by subtracting neuropil fluorescence from raw fluorescence ($F_{ROI}$ = $F_{raw}$ − 0.7*$F_{neuropil}$). We used the *ksdensity* function to determine $F_0$ and calculate $\Delta F/F_0$ (($F$ − $F_0$)/$F_0$). The following analyses were done in Python: firing rates, maximum response magnitude, correlation coefficients, encoding models, and decoding analysis. The reliability indices of single neurons in response to natural movies were calculated by averaging the correlation of all pairwise combinations of trials for a single movie (*Rikhye and Sur, 2015*). For computation of firing rates and pairwise correlation coefficients, we used deconvolved spikes from Suite2p as an indirect readout of spiking activity of individual neurons. For pairwise neuron-to-neuron correlation, each neuron's activity for all trials was concatenated into one vector. Pearson correlation coefficients were computed between a pair of vectors using *numpy.corrcoef* function in Python. For signal and noise correlations, only stimulus-dependent responses of individual neurons were used. For signal correlation, each neuron's responses to a set of stimuli were averaged across trials and the vector containing trial-average activity was used for Pearson correlation. For noise correlation, z-scores were computed across the trials, which were then concatenated into a single vector for Pearson correlation.

## Pupil dynamics

For pupil tracking, we used DeepLabCut (*Mathis et al., 2018*; *Nath et al., 2019*) (version 3.0.0). Specifically, we manually labeled ~200 frames (14 frames/video) and used a mobilenet_v2_1.0-based neural network with default parameters for 50,000 training iterations to predict the location of 8 markers (*xy*-coordinates). Once the network was trained, it was used to place coordinates on unlabeled frames. This network was then used to analyze videos from similar experimental settings. The pupil diameter was calculated by computing the distance between pairs of xy-coordinates placed across the pupil.

## GLM of neuronal activity

Neuronal encoding of (1) visual stimulus, pupil diameter, and running speed and (2) population activity was modeled with a GLM for each neuron independently. A GLM with linear (identity) link function was used to compute the weights of predictors in modeling the activity of single neurons based on calcium signals. In this model, neuronal activity is described as a linear sum of visual and behavioral predictors aligned to each event. For the first encoding model, predicted neuronal activity $r_n(t)$ for a target neuron $n$ is described as

$$r_n(t) = \sum_c \sum_{t_s \in S_c} \beta_{c,n}^{t_s} x_c(t - t_s) + \sum_b \beta_{b,n} x_b(t) + \varepsilon,$$

where $c$ represents the direction of the visual stimulus (eight directions), $b$ represents the behavioral variables (pupil diameters and running speed), $S_c$ represents the set of times to cover each predictor window. $\beta_{c,n}$ and $\beta_{b,n}$ represent the weights of visual stimulus and behavioral variables for neuron $n$. The visual stimulus predictors cover the window [0–2] s from stimulus onset. $x_c$ and $x_b$ represent the visual stimulus and behavioral variable predictors, respectively. Each predictor is coded as '1' or '0' except for behavioral variables. Behavioral variable predictors are continuous behavioral event variables such as pupil diameter. $\varepsilon$ is the model bias (intercept). The values for the behavioral variables were *z*-scored.

To estimate the optimal weights for each neuron without overfitting, the *lassoglm* function in MATLAB with 10-fold cross-validation of the training set was used with a lasso regularization according to the value of a selected parameter $\lambda$, which represents regularization coefficients (*sklearn.linear_model.Lasso* class in Python was used for the population activity encoding model). The value of $\lambda$ in the lassoglm function was set to be $10^{-3}$. Model performance was assessed for the test dataset by quantifying explained variance ($R^2$).

To determine the proportion of neurons with a significant contribution of the variance explained by each variable, we fitted the model using full predictors (full model) and predictors in which the target predictor is set to zero within whole-time points (partial model) and calculated the explained variance ($R^2_{full}$, $R^2_{partial}$) for the full and partial model. We then performed a *t*-test between the $R^2_{full}$ and $R^2_{partial}$ and across the 10-fold cross-validation, correcting for multiple comparisons with Bonferroni–Holm correction.

## Decoding analysis

We built SVM classifiers using *sklearn.svm.SVC* class in Python to evaluate the capacity of the V1 neuronal population to represent visual stimuli information. All population decoding was performed for each session. For the training and testing dataset, the number of trials in each condition (eight directions or seven movies) was matched to prevent bias for training classifiers. We left a 33% subset of trials for prediction to avoid overfitting. Best estimator parameters were determined by optimization to avoid overfitting and minimize loss of validation in a grid search manner (search range $10^{-3}$ to $10^3$).

We used the mean DF/F response during stimulus presentation (2 s for gratings, 3 s for movies) for each individual neuron. Classifier performance on each iteration was estimated by averaging decoding accuracies across all iterations. To determine if the decoder performance was above chance, we shuffled labels for the test data, trained, and tested the decoder to assess the decoding accuracy.

For statistical assessment of the decoding accuracy between sessions, we trained and tested decoders using a subset of population of neurons (from 5 to 25 neurons, at increments of 5) by randomly choosing neurons in each iteration. The decoding performance was evaluated by calculating

the AUROC from prediction scores of the test set. Difference in decoding accuracy between the two groups (control and Gat3 KO) was evaluated using a two-way ANOVA to examine the effects of population size and experimental group on the decoding accuracy.

## Statistics

All statistical analysis was performed using custom written scripts in MATLAB or Python. All statistical tests used are reported in the figure legends. For comparison of single neuron responses collected *in vivo*, we used linear mixed effects models (referred to as LME) to accommodate the dependency between measurements taken from the same subject (*Aarts et al., 2014*) using *statsmodel.formula. api.mixedlm* from statsmodel (*Seabold and Perktold, 2010*) (version 0.14.4). Statistical comparisons employed two-tailed *t*-statistics. For *ex vivo* and *in vitro* experiments, we used Mann–Whitney *U* tests, Kolmogorov–Smirnov tests, and two-tailed unpaired *t*-tests as appropriate. For SVM decoding, we used a two-way ANOVA and Mann–Whitney *U* test.

## Acknowledgements

We thank Taylor Johns for lab management and members of the Sur laboratory for many discussions and comments. We are grateful to Alexandria Barlowe for histology and animal colony management. This work was supported by NIH grants R01DA049005, R01MH126351, R01NS130361, and R01MH133066, MURI Grant W911NF2110328, and The Picower Institute Innovation Fund (MS); NIH Fellowship F32EY022264 (GS); Simons Foundation Autism Research Initiative Bridge to Independence award (XT); and a fellowship from The JPB Foundation (GF).

## Additional information

### Funding

| Funder | Grant reference number | Author |
| --- | --- | --- |
| National Institutes of Health | R01DA049005 | Mriganka Sur |
| National Institutes of Health | R01MH126351 | Mriganka Sur |
| National Institutes of Health | R01NS130361 | Mriganka Sur |
| National Institutes of Health | R01MH133066 | Mriganka Sur |
| United States Department of Defense | Multidisciplinary University Research Initiative W911NF2110328 | Mriganka Sur |
| The Picower Institute Innovation Fund | | Mriganka Sur |
| National Institutes of Health | F32EY022264 | Grayson O Sipe |
| Simons Foundation | Autism Research Initiative Bridge to Independence award | Xin Tang |
| JPB Foundation | Picower Postdoctoral Fellowship | Giselle Fernandes |

The funders had no role in study design, data collection, and interpretation, or the decision to submit the work for publication.

### Author contributions

Jiho Park, Conceptualization, Data curation, Formal analysis, Validation, Investigation, Visualization, Methodology, Writing – original draft, Writing – review and editing; Grayson O Sipe, Xin Tang,

Conceptualization, Funding acquisition, Investigation, Methodology, Writing – review and editing; Prachi Ojha, Giselle Fernandes, Investigation, Visualization, Methodology; Yi Ning Leow, Formal analysis, Writing – review and editing; Caroline Zhang, Arundhati Natesan, Gabrielle T Drummond, Investigation; Yuma Osako, Methodology; Rudolf Jaenisch, Conceptualization, Supervision, Funding acquisition; Mriganka Sur, Conceptualization, Resources, Supervision, Funding acquisition, Project administration, Writing – review and editing

### Author ORCIDs
Jiho Park ⓘ https://orcid.org/0009-0003-9533-7107
Giselle Fernandes ⓘ https://orcid.org/0000-0003-1388-6392
Yi Ning Leow ⓘ https://orcid.org/0000-0003-4600-5918
Mriganka Sur ⓘ https://orcid.org/0000-0003-2442-5671

### Ethics
All experimental procedures performed in this study were approved by the Massachusetts Institute of Technology's Animal Care and Use Committee (protocol #: 2308-000-562) and conformed to the National Institutes of Health.

Reviewer #1 (Public review): https://doi.org/10.7554/eLife.107298.3.sa1
Reviewer #2 (Public review): https://doi.org/10.7554/eLife.107298.3.sa2
Author response https://doi.org/10.7554/eLife.107298.3.sa3

---

## Additional files

### Supplementary files
MDAR checklist

### Data availability
All data generated or analyzed during this study are included in the manuscript; data and code are shared in GitHub (copy archived at *Park, 2024*).

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
