## [Editor Report · eLife Assessment]

In this manuscript, Park et al. developed a multiplexed CRISPR construct to genetically ablate the GABA transporter GAT3 in the mouse visual cortex, with effects on population-level neuronal activity. This work is **important**, as it sheds light on how GAT3 controls the processing of visual information. The findings are **compelling**, leveraging state-of-the-art gene CRISPR/Cas9, in vivo two-photon laser scanning microscopy, and advanced statistical modeling.

---

## [Referee Report · Reviewer #1 (Public review)]

Summary:

The authors have investigated the role of GAT3 in the visual system. First, they have developed a CRISPR/Cas9-based approach to locally knock out this transporter in the visual cortex. They then demonstrated electrophysiologically that this manipulation increases inhibitory synaptic input into layer 2/3 pyramidal cells. They further examined the functional consequences by imaging neuronal activity in the visual cortex in vivo. They found that absence of GAT3 leads to reduced spontaneous neuronal activity and attenuated neuronal responses and reliability to visual stimuli, but without an effect on orientation selectivity. Further analysis of this data suggests that Gat3 removal leads to less coordinated activity between individual neurons and in population activity patterns, thereby impaired information encoding. Overall, this is an elegant and technically advanced study that demonstrates a new and important role of GAT3 in controlling processing of visual information.

Strengths:

Development of a new approach for a local knockout (GAT3)

Important and novel insights into visual system function and its dependence on GAT3

Plausible cellular mechanism

Weaknesses:

No major weaknesses.

---

## [Referee Report · Reviewer #2 (Public review)]

Summary:

Park et al. has made a tool for spatiotemporally restricted knockout of the astrocytic GABA transporter GAT3 leveraging CRISPR/Cas9 and viral transduction in adult mice, and evaluated the effects of GAT3 on neural encoding of visual stimulation.

Strengths:

This concise manuscript leverages state-of-the-art gene CRISPR/Cas9 technology for knocking out astrocytic genes. This has to a little degree been preformed previously in astrocytes and represents an important development in the field. Moreover they utilize in vivo two-photon imaging of neural responses to visual stimuli as a readout of neural activity, in addition to validating their data with ex vivo electrophysiology. Lastly, they use advanced statistical modeling to analyze the impact on GAT3 knockout. Overall, the study comes across as rigorous and convincing.

Weaknesses:

Adding the following experiments would potentially have strengthened the conclusions and helped interpret the findings, although may be considered outside the scope of this manuscript, and be pursued in future work:

(1) Neural activity is quite profoundly influenced by GAT3 knockout. Corroborating these relatively large changes to neural activity with in vivo electrophysiology of some sort as an additional readout would have strengthened the conclusions.

(2) Given the quite large effects on neural coding in visual cortex assessed with jRGECO imaging it would have been interesting the mouse groups could have been subjected to behavioral testing assessing the visual system.

---

## [Author Response]

The following is the authors’ response to the original reviews.

**Reviewer #1 (Public review):**
Summary:The authors have investigated the role of GAT3 in the visual system. First, they have developed a CRISPR/Cas9-based approach to locally knock out this transporter in the visual cortex. They then demonstrated electrophysiologically that this manipulation increases inhibitory synaptic input into layer 2/3 pyramidal cells. They further examined the functional consequences by imaging neuronal activity in the visual cortex in vivo. They found that the absence of GAT3 leads to reduced spontaneous neuronal activity and attenuated neuronal responses and reliability to visual stimuli, but without an effect on orientation selectivity. Further analysis of this data suggests that Gat3 removal leads to less coordinated activity between individual neurons and in population activity patterns, thereby impairing information encoding. Overall, this is an elegant and technically advanced study that demonstrates a new and important role of GAT3 in controlling the processing of visual information.

We are grateful to the reviewer for their positive appraisal of our work, including our technical advances and our demonstration of how cortical astrocytes play a role in visual information processing by neurons via GAT3-mediated regulation of activity.

Strengths:(1) Development of a new approach for a local knockout (GAT3).(2) Important and novel insights into visual system function and its dependence on GAT3.(3) Plausible cellular mechanism.Weaknesses:No major weaknesses were identified by this reviewer.

We thank the reviewer for highlighting the strengths of our study, including the development of a novel local knockout strategy for GAT3, the discovery of important functional consequences for visual system processing, and the identification of a plausible underlying cellular mechanism.

**Reviewer #2 (Public review):**
Summary:Park et al. have made a tool for spatiotemporally restricted knockout of the astrocytic GABA transporter GAT3, leveraging CRISPR/Cas9 and viral transduction in adult mice, and evaluated the effects of GAT3 on neural encoding of visual stimulation.Strengths:This concise manuscript leverages state-of-the-art gene CRISPR/Cas9 technology for knocking out astrocytic genes. This has only to a small degree been performed previously in astrocytes, and it represents an important development in the field. Moreover, the authors utilize in vivo two-photon imaging of neural responses to visual stimuli as a readout of neural activity, in addition to validating their data with ex vivo electrophysiology. Lastly, they use advanced statistical modeling to analyze the impact of GAT3 knockout. Overall, the study comes across as rigorous and convincing.

We appreciate the reviewer’s endorsement of our experimental rigor and methodological innovation. We agree that combining *in vivo* and *ex vivo* measurements with rigorous analytical methods strengthens the overall conclusions of the study and demonstrates the important role of astrocytic GAT3 in cortical visual processing.

Weaknesses:Adding the following experiments would potentially have strengthened the conclusions and helped with interpreting the findings:(1) Neural activity is quite profoundly influenced by GAT3 knockout. Corroborating these relatively large changes to neural activity with in vivo electrophysiology of some sort as an additional readout would have strengthened the conclusions.

We agree that further investigation of neuronal activity at higher temporal resolution would provide valuable complementary data, particularly given the profound effects we observed using a pan-neuronal calcium indicator. Detailed *in vivo* electrophysiology—such as large-scale Neuropixel recordings—would allow assessment of single-neuron spiking dynamics and potentially cell-type specific responses following GAT3 deletion. While such an investigation is beyond the scope of the current study, we concur that it would be an important follow-up direction to further dissect the effects of GAT3 knockout on neuron activity profiles at both single-cell and population levels.

(2) Given the quite large effects on neural coding in visual cortex assessed på jRGECO imaging, it would have been interesting if the mouse groups could have been subjected to behavioral testing, assessing the visual system.

We appreciate the reviewer’s suggestion to explore potential behavioral consequences of GAT3 deletion. Based on our observed alterations in visual cortical activity, we agree that GAT3 knockout could impact visual discrimination-based behaviors. Astrocytes in the visual cortex are highly tuned to sensory and motor events and are generally known to shape behavioral outputs (Slezak et al., 2019; Kofuji & Araque, 2021). Our study suggests that regulation of inhibitory signaling via GAT3 transporters is a possible mechanism by which astrocytes influence visually guided behaviors. Although behavioral assessments fall beyond the scope of the current work, we agree with the reviewer’s suggestion and will pursue future experiments employing paradigms such as go/no-go visual detection or two-alternative forced choice to determine whether astrocytic GAT3 modulates visually guided behaviors and perceptual decisionmaking.

**Reviewer #1 (Recommendations for the authors):**
It could be more clearly stated from the very beginning that a method was developed and used which, by itself, apparently has no cell type selectivity. It is highly plausible that the effects are mostly due to the absence of astrocytic GAT3, as discussed by the authors, but the distinction of what has been done and what is interpretation based on the literature is occasionally a bit blurry. This is also important because there are CRISPR/Cas9-based approaches that are astrocyte-specific (e.g., GEARBOCS).

We thank the reviewer for this helpful suggestion. As noted, our current approach does not confer celltype specificity on its own. Although our interpretation—supported by expression patterns and prior literature—attributes the observed effects primarily to astrocytic GAT3 loss, we agree that this distinction should be explicitly stated. We have revised the Introduction section (lines 83-87) to clarify that while MRCUTS allows for local gene knockout, it is not inherently cell-type specific unless combined with celltype restricted Cre drivers, as is possible in future applications.

A change of ambient GABA following GAT3 deletion is central to the proposed cellular mechanism. Demonstrating this directly would strengthen the manuscript (e.g., changed tonic GABAergic current in the absence of GAT3, and insensitivity to SNAP-5114).

While we recognize that directly quantifying ambient GABA levels would further strengthen our study, substantial evidence supports the role of GABA transporters in coordinately regulating both phasic and tonic inhibition and cellular excitability (Kinney, 2005; Keros & Hablitz, 2005; Semyanov et al. 2003).

Moreover, tonic GABA currents have been shown to strongly correlate with phasic inhibitory bursts (Glykys & Mody, 2007; Farrant & Nusser, 2005; Ataka & Gu, 2006), suggesting shared underlying regulatory mechanisms. Furthermore, as the reviewer correctly points out, alternative mechanisms such as non-vesicular GABA release or disinhibition via interneuron suppression cannot be excluded (also discussed in Kinney 2005). Given these considerations, we prioritized sIPSC measurements as a more integrative and reliable proxy for altered GABAergic signaling in L2/3 pyramidal neurons. We have revised the Discussion section (lines 329-333) to explain our choice of approach for further clarification.

We also agree it would be of interest to test whether GAT3 KO neurons exhibit insensitivity to SNAP-5114, both *ex vivo* and *in vivo*. However, based on our SNAP-5114 application experiments *in vivo*, which revealed only subtle effects on single-neuron properties (Figure S2A-F), we anticipate that interpreting a lack of effect in the KO condition would be challenging and potentially inconclusive.

References

Ataka, T. & Gu, J. G. Relationship between tonic inhibitory currents and phasic inhibitory activity in the spinal cord lamina II region of adult mice. Mol. Pain. (2006).

Bright, D. & Smart, T. Methods for recording and measuring tonic GABAA receptor-mediated inhibition. Front. Neural Circuits. 7, (2013).

Farrant, M. & Nusser, Z. Variations on an inhibitory theme: phasic and tonic activation of GABAA receptors. Nat. Rev. Neurosci. 6, 215–229 (2005).

Glykys, J. & Mody, I. Activation of GABAA Receptors: Views from Outside the Synaptic Cleft. Neuron. 56, 763-770 (2007).

Keros, S. & Hablitz, J. J. Subtype-Specific GABA Transporter Antagonists Synergistically Modulate Phasic and Tonic GABAA Conductances in Rat Neocortex. J. Neurophysiol. 94, 2073–2085 (2005).

Kinney, G. A. GAT-3 Transporters Regulate Inhibition in the Neocortex. J. Neurophysiol. 94, 4533–4537 (2005).

Kofuji, P. & Araque, A. Astrocytes and Behavior. Annu. Rev. Neurosci. 44, 49–67 (2021).

Semyanov, A., Walker, M. & Kullmann, D. GABA uptake regulates cortical excitability via cell type–specific tonic inhibition. Nat. Neurosci. 6, 484–490 (2003).

Slezak, M., Kandler, S., Van Veldhoven, P. P., Van den Haute, C., Bonin, V. & Holt, M.G. Distinct

Mechanisms for Visual and Motor-Related Astrocyte Responses in Mouse Visual Cortex. Curr. Biol. 18, 3120-3127 (2019).